Two continents and two names for a Neotropical colletid bee species (Hymenoptera: Colletidae: Neopasiphaeinae): Hoplocolletes ventralis (Friese, 1924)

Almeida Eduardo A.B. eduardo@ffclrp.usp.br
Quinteiro Fábio B.
Departamento de Biologia—FFCLRP, Universidade de São Paulo , Ribeirão Preto, SP , Brazil
Hedrick Ann
Electronic publication date: 2015 Oct 20
Publication date: 2015
Volume: 3
Electronic Location ID: e1338
Received 2015 Aug 15; Accepted 2015 Sep 29
Copyright: © 2015 Almeida and Quinteiro
Copyright year: 2015
Copyright holder: Almeida and Quinteiro
License: This is an open access article distributed under the terms of the Creative Commons Attribution License, which permits unrestricted use, distribution, reproduction and adaptation in any medium and for any purpose provided that it is properly attributed. For attribution, the original author(s), title, publication source (PeerJ) and either DOI or URL of the article must be cited.
License URL: https://creativecommons.org/licenses/by/4.0/

Keywords: Australia, Taxonomy, Apoidea, Biogeography, Systematics, Brazil

Funding: São Paulo Research Foundation (FAPESP) 2011/09477-9 Brazilian Council for Scientific and Technological Development, CNPq 142211/2012-5 This project was supported by grant 2011/09477-9, São Paulo Research Foundation (FAPESP) to EAB Almeida; FB Quinteiro is supported by a PhD fellowship granted by the Brazilian Council for Scientific and Technological Development, CNPq 142211/2012-5. The funders had no role in study design, data collection and analysis, decision to publish, or preparation of the manuscript.

==============================
Neopasiphaeine bees (Apoidea: Colletidae) are known for their Amphinotic distribution in the Australian and Neotropical regions. Affinities between colletid taxa in Australia and South America have been speculated for decades, and have been confirmed by recent phylogenetic hypotheses that indicate a biogeographic scenario compatible with a trans-Antarctic biotic connection during the Paleogene. No neopasiphaeine species occurs on both sides of the Pacific Ocean, but the Neotropical species Hoplocolletes ventralis (Friese, 1924) was described as an Australian taxon due to an error in the specimen labels. This mistake was recognized by CD Michener 50 years ago. We herein report that the same labeling problem also happened with Dasycolletes chalceus Friese, 1924, which remained as a tentatively placed species in the Australian genus Leioproctus until now. Moreover, Dasycolletes chalceus is interpreted as a synonym of Hoplocolletes ventralis. We also provide a revised diagnosis for Hoplocolletes, describe the male of H. ventralis in detail for the first time, including a comparative study of its genitalia and associated sterna.

Introduction

Affinities between taxa of Colletidae distributed in Australia and South America have been speculated for decades (Michener, 1965; Michener, 1989), and have been confirmed by recent phylogenetic hypotheses that indicate a biogeographic scenario compatible with a trans-Antarctic biotic connection during the Paleogene (Almeida et al., 2012). Dasycolletes ventralis Friese, 1924 was described as an Australian colletid species based on a single female specimen labeled as having been collected in Sydney (Australia). The species actually is endemic to Brazil, and there is no species occurring in Australia that could be confused with it. The confusion certainly results from an error in the label, as concluded by Michener (1965: p. 41), an interpretation followed by subsequent authors (e.g., Moure, Graf & Urban, 2007; Rasmussen & Ascher, 2008). After the species description, it was moved to the genus Paracolletes by Cockerell (1929), and later placed in Leioproctus (Hoplocolletes), created by Michener (1965) to accommodate it based on clear affinities to other taxa classified as Leioproctus, but also recognizing its uniqueness (see also Michener, 1989; Michener, 2007). Hoplocolletes remains a monotypic taxon in Neopasiphaeinae (Colletidae), having been classified as genus (e.g., Silveira, Melo & Almeida, 2002; Moure, Graf & Urban, 2007; Almeida & Danforth, 2009; Almeida et al., 2012) or as subgenus of Leioproctus (e.g., Michener, 1965; Michener, 1989; Michener, 2007), the former being followed in this paper.

Hoplocolletes ventralis has been recorded in three states in southeastern Brazil: Espírito Santo, Minas Gerais, Rio de Janeiro (Silveira, Melo & Almeida, 2002; Moure, Graf & Urban, 2007). Nevertheless, it remains a poorly known genus, with relatively little distributional information, the male undescribed, host-plant preferences unknown, and the only piece of bionomical information for this species is that it is a soil nesting bee (EAB Almeida, pers. obs., 2001). The phylogenetic affinities of Hoplocolletes and other neopasiphaeine taxa were uncertain until molecular phylogenetic hypotheses placed this taxon in a clade comprising Eulonchopria and Nomiocolletes (Almeida & Danforth, 2009; Almeida et al., 2012). Michener (1989: p. 630) suggested that Hoplocolletes could be part of a “Basal Group,” characterized by the fully developed sternal scopa. Based on the phylogenetic hypotheses currently available, it seems that this scopa arose multiple times in the Neopasiphaeine clade, since taxa with this character, Hoplocolletes, Cephalocolletes, Reedapis, and Tetraglossula are otherwise not close relatives (Almeida & Danforth, 2009; Almeida et al., 2012).

The aim of this work is three fold. To resolve a taxonomic problem related to a new synonymy involving Hoplocolletes ventralis and Dasycolletes chalceus, which are here interpreted as synonyms. To report that the above mentioned labeling problem that made the taxonomic history of Hoplocolletes ventralis problematic also happened with Dasycolletes chalceus Friese, 1924, which remained as a tentatively placed species in the Australian genus Leioproctus until now (Michener, 1965; Cardale, 1993; Almeida, 2008; Rasmussen & Ascher, 2008). To increase the knowledge about the morphology and distinctiveness of Hoplocolletes, particularly by providing a novel description of the male genital complex for this species.

Material & Methods

Part of the material studied is deposited in the Entomological Collection “Prof. JMF Camargo” [RPSP] in Departamento de Biologia (FFLRP/USP, Ribeirão Preto, Brazil). A male specimen of Hoplocolletes ventralis was obtained on loan from Entomological Collection “Pe. JS Moure” [DZUP], Departamento de Zoologia (UFPR, Curitiba, Brazil), and the female type specimen of Dasycolleletes chalceus Friese, 1924 was studied and photographed at the entomological collection of Museum für Naturkunde [ZMB] (Berlin, Germany). Photographs of the female specimen of Dasycolletes ventralis Friese, 1924, deposited at the American Museum of Natural History (AMNH) collection, were kindly made available for this study.

The general morphological terminology follows Michener (2007). Antennal flagellomeres are indicated as F1, F2, etc.; metasomal terga and sterna, respectively, as T1 to T7, and S1 to S8. The density of punctation and intervals between the punctures are based on relative puncture diameter, pd (e.g., <1 pd: less than 1× the puncture diameter between the punctures). Color images were obtained on a Zeiss Axiocam 206 color camera associated to a Zeiss Discovery. V12 stereomicroscope, or with an AmScope MU1000A Digital Camera adapted onto a Leica MZ6 stereomicroscope; pictures were assembled with the software Helicon Focus 6.2.

Results

The species Dasycolletes chalceus was not studied after its original description. It was described in the same publication and same page as Dasycolletes ventralis (Friese, 1924: p. 218). After 1924, it was only mentioned in catalogues and revisionary works (e.g., Michener, 1965; Cardale, 1993; Almeida, 2008; Rasmussen & Ascher, 2008), but the type specimen was never studied again. The only exemplar of Dasycolletes chalceus located and bearing Friese’s original labels is deposited in ZMB (Fig. 1). It clearly has all diagnostic characters for Hoplocolletes as currently circumscribed, and no differences were found in relation to Hoplocolletes ventralis either. Hence, they are herein synonymized. The only known specimen of Dasycolletes ventralis bearing Friese’s original labels is in the American Museum of Natural History collection (New York, USA) (Fig. 2) and it is the same female studied by Michener (1965) that lead him to conclude that it was not an Australian taxon, as indicated by the collecting labels, but a specimen probably collected in Brazil. The interpretation of Friese’s types is a controversial subject and it is likely that the AMNH specimen is a duplicate, not the primary type (Rasmussen & Ascher, 2008; JS Ascher, pers. comm., 2015). But, so far, it is the only specimen labeled by Friese himself as Dasycolletes ventralis available for study. It is worth noting that both specimens were probably collected together, have locality labels that are identical, “Australia ∖∖ Sydney ∖∖ 14.9/06.” The collector’s name is lacking from the D. chalceus specimen label but is in the species’ description (Friese, 1924: p. 218): “von Sydney im September, Frank leg.”

Hoplocolletes ventralis (Friese, 1924)

Dasycolletes ventralis Friese, H. (1924) [218].	
Type data: syntype AMNH 〈F〉.	
Type locality: ‘Australia, Sydney’.	
Dasycolletes chalceus Friese, H. (1924) [218], new synonymy.	
Type data: syntype ZMB 〈F〉.	
Type locality: ‘Australia, Sydney’.	

Description of male: Approximate body length: 10 mm; length of forewing: 7.7 mm; maximum width of metasoma (T2): 2.5 mm. Color: predominantly black; apical half of mandible, ventral surface of F2–F11, tibiae, femora, trochanters, S2–S3, apical margins of terga dark reddish brown. Tarsi light brown. Tegula, pterostigma and wing veins dark brown; wing membrane brown infumated. Pubescence: predominantly pale yellowish or cream on entire body. Face and pronotal lobe with abundant pubescence; clypeus with decumbent to semidecumbent pilosity (0.5 mm in length), more erect and shorter on paraocular area and frons (0.3–0.45 mm in length). Mesoscutum with scarce pilosity. Lateral pilosity of mesepisternum semidecumbent and sparse (0.25–0.35 mm in length). Integumental surface: punctation coarse and dense on clypeus (≤1 pd), finer and denser frons (<1 pd), on vertex variable (denser medially, sparser (≤1 pd) laterally as well as on gena) integument smooth and shiny between punctures; coarse and dense on mesosoma, sparser toward center of disc of mesoscutum, and inferior on mesepisternum; metapostnotum smooth and shiny, delimited from pronotum by a pit-row; T1 smooth and shiny, with very sparse (2–7 pd) moderately coarse punctation, transversal line of barely aligned punctures delimiting marginal region of T1; on T2 slightly denser than on T1, but punctation leaving broad shiny areas as well; T3 and T4 with basal portion finely and densely punctated, sparser and coarser distad. Structure (measurements in mm): head about 1.1× wider than long (2.66:2.43); inner orbits converging below (upper to lower interorbital distance, 1.76:1.47), inner margin almost straight; eye about 3.6× longer than its maximum width in frontal view (1.76:0.48), in lateral view about 1.2× wider than gena (0.74:0.64). Vertex well developed above ocelli (distance between upper margin of lateral ocellus and vertex = 0.53), comparable to ocelloocular distance (0.51); interocellar distance = 0.14; diameter of median ocellus = 0.25. Approximate length of antenna = 4.0, length and maximum width of scape = 0.73, 0,2; of pedicel = 0.16; of F1 = 0.19; F2 about 1.5× wider than long (0.18:0.27); F3 about 1.5× longer than wide (0.30:0.21). Mesoscutum length = 1.83, intertegular distance = 1.75. Genital capsule and male S7 and S8 as illustrated in Figs. 4 and 5 (see discussion about the male terminalia below, in the ‘Revised Diagnosis’ for Hoplocolletes).

Figure 1 Female specimen of Dasycolletes chalceus Friese, 1924 deposited at the Museum für Naturkunde collection [ZMB] (photo credit: Eduardo A.B. Almeida).

(A) Dorsal habitus (scale bar = 1 mm), (B) lateral habitus, (C) face, (D) ventral metasomal scopa, (E) labels.

Figure 2 Female specimen of Dasycolletes ventralis of Friese, 1924 deposited at the American Museum of Natural History collection [AMNH] (photo credit: Hadel Go).

(A) Dorsal habitus, (B) lateral habitus, (C) face, (D) magnified view of ventral metasomal scopa, (E) labels.

Figure 3 Male specimen of Hoplocolletes chalceus (Friese, 1924) from Itapina, ES, Brazil [DZUP] (photo credit: Eduardo A.B. Almeida).

(A) Lateral habitus, (B) dorsal habitus, (C) face, (D) mesosoma and anterior metasoma; scale bars = 1 mm.

Figure 4 Comparative morphology of male metasomal sterna S7 and S8 (dorsal views shown on left) of Hoplocolletes ventralis (Friese, 1924) and related neopasiphaeine taxa.

Comparative morphology of male metasomal sterna S7 and S8 (dorsal views shown on left) of Hoplocolletes ventralis (Friese, 1924), Nomiocolletes joergenseni (Friese, 1908), and Reedapis semicyanea (Spinola, 1851). ALb, apical lobe of S7; BLb, basal lobe of S7; LLb, lateral lobe of S7; LPr, lateral process of S8; MPr, median process of S8; scale bars = 0.5 mm. Cladogram represents a hypothesis for the phylogenetic relationships among these three taxa (Almeida & Danforth, 2009).

Figure 5 Comparative morphology of male genitalia (dorsal views shown on left) of Hoplocolletes ventralis (Friese, 1924) and related neopasiphaeine taxa.

Comparative morphology of male genitalia (dorsal views shown on left) of Hoplocolletes ventralis (Friese, 1924), Nomiocolletes joergenseni (Friese, 1908), and Reedapis semicyanea (Spinola, 1851). ApP, apodeme of penis valve; Cs, cuspis of volsella; Dg, digitus of volsella; Gbs, gonobase; Gcx, gonocoxa; Gns, gonostyle; PV, penis valve; SPV, ventral spine of penis valve; scale bars = 0.5 mm. Cladogram represents a hypothesis for the phylogenetic relationships among these three taxa (Almeida & Danforth, 2009).

Hoplocolletes Michener, 1965

Revised diagnosis for the genus (characters apply to both sexes unless otherwise stated). Length 10–12 mm. Body black to dark brown; head and mesosoma with conspicuous coarse punctation; T1 and T2 largely impunctate, smooth and shining (remaining terga rather finely and closely punctate); pubescence short, sparse, blackish to dark brown on female (light yellow to fulvous on male), except on hind legs and metasomal sterna where there are long, pale hairs; metasomal hair bands absent, male clypeus with plumose and semidecumbent pubescence. Mandible with an ordinary preapical tooth. Inner orbits subparallel (female, Figs. 1D and 2C) or converging below (male, Fig. 3C). Facial fovea absent; clypeus weakly convex; labrum with apical margin concave medially, elevated zone highest medially, occupying about basal half of labrum (Michener, 1989: Fig. 7Q). Preoccipital carina absent; malar area linear; clypeus little protuberant. Male flagellum elongate (approximately 3.0 mm long), F2 longer than wide. Vertex produced behind ocelli and eyes (Figs. 1A, 1D, 2A, 2C, 3A, 3B and 3C). Apex of scape of female reaching upper margin of median ocellus (Figs. 1D and 2C); antennae arising about middle of face. Dorsolateral angle of pronotum low, rounded, scarcely evident; metapostnotum smooth, marginal line pitted, its basal part slightly longer than metanotum. Femoral scopa sparse, formed by long delicate branched hairs, those behind corbicula and on trochanter long but simple; tibial scopal hairs dividing to form few major branches. Female basitibial plate distinct, hairs short, appressed, different from those of adjacent areas, marginal carinae clearly exposed. Inner hind tibial spur of female coarsely pectinate with 5–6 teeth (Michener, 1989: Fig. 7Q). Forewing with three submarginal cells, second much shorter than third and receiving recurrent vein beyond middle (Figs. 1F and 3A); basal vein of forewing meeting cu-v (Fig. 1); stigma large, long, not quite parallel sided, two-thirds as long as costal side of marginal cell, marginal cell longer than distance between its apex and wing apex. T1 dorsally approximately twice wider than long; S3–S5 of female with dense, long (shorter than exposed part of sternum), pale yellow, simple hairs (some hooked at tips) forming band occupying apical half of each sternum, female S2 with similar but sparser hair band (Figs. 1C and 2D); S3–S5 of male with a longer hairs near apical margin, S5 with distinct apical fringe.

Male genital capsule and associated sterna of Hoplocolletes ventralis are illustrated in Figs. 5 and 6 along with exemplar species of two other neopasiphaeine genera: Nomiocolletes joergenseni (Friese, 1908) and Reedapis semicyanea (Spinola, 1851). According to the phylogenetic hypotheses of Almeida & Danforth (2009) and Almeida et al. (2012), Hoplocolletes and Nomiocolletes are closely related lineages, whereas Reedapis is part of a more distantly related clade. The interpretation of homologies for the male terminalia in this comparative context makes the understanding of relevant characters of Hoplocolletes more defensible. Apical process of male S7 comprising two lobes on each side: one apicolateral more developed and hairier (Fig. 5: lateral lobe—LLb) the other closer to the base of this process (Fig. 5: basal lobe—BLb), Nomiocolletes is distinctive in having a bilobed lateral lobe; Hoplocolletes does not have apical protuberances as found in other Neopasiphaeinae (Fig. 5: apical lobe—ALb); apodeme of S7 relatively long in relation to the apical process. Median process of male S8 (Fig. 5: MPr) similar in length to the remainder of S8; spiculum ordinary (not as produced as in Nomiocolletes or Reedapis). Gonobase of male genitalia (Fig. 6: Gbs) less than 1/4 of total length of genital capsule; gonostylus and gonocoxite (Fig. 6: Gns, Gcx) fused on dorsal surface but separable on ventral surface, apex of gonostylus rounded and not bent ventrad (directed mesad and ventrad in Nomiocolletes and Reedapis); apex of penis valve (Fig. 6: PV) bent ventrad, ventral spine well-developed (Fig. 6: SPV).

We are grateful to Gabriel A.R. Melo (Universidade Federal do Paraná—[DZUP]) who kindly provided information about a locality where Hoplocolletes ventralis is known to occur in Minas Gerais state, and loaned a male specimen used in this study. We are also indebted to Frank Koch for hosting EAB Almeida during a visit to the Museum für Naturkunde [ZMB], and to Hadel H. Go and Jerome G. Rozen, Jr (American Museum of Natural History—AMNH) for kindly taking photographs and making them available for this work. Our thanks to John S. Ascher (National University of Singapore) and Gabriel A.R. Melo for valuable discussion on the systematics of neopasiphaeine bees and the work of H Friese, and to Connal Eardley, Diego S. Porto, and one anonymous reviewer for critically commenting on this manuscript.

Additional Information and Declarations

Competing Interests

Author Contributions

The authors declare there are no competing interests.

Eduardo A.B. Almeida conceived and designed the experiments, performed the experiments, analyzed the data, contributed reagents/materials/analysis tools, wrote the paper, prepared figures and/or tables, reviewed drafts of the paper.

Fábio B. Quinteiro performed the experiments, analyzed the data, prepared figures and/or tables, reviewed drafts of the paper.

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
