# Peer review of "Two continents and two names for a Neotropical colletid bee species (Hymenoptera: Colletidae: Neopasiphaeinae): Hoplocolletes ventralis (Friese, 1924)"

_PeerJ, doi:10.7717/peerj.1338_

## Round 0.1 · original submission · Minor Revisions

· Academic Editor

Minor Revisions

Please revise according to the reviewers' comments.

·

Basic reporting

I understand that the article meets the Journal's standard.

Experimental design

I understand that the article meets the Journal's standard.

Validity of the findings

I think the finding are perfectly valid.

Comments for the author

The article is interesting and absolutely necessary to resolve a few taxonomic issues. It is well presented and the figures are excellent. I have a few suggestions that I believe will make it easier for readers to follow. They are included in the manuscript.

Reviewer 2 ·

Basic reporting

The article is clearly written, self-contained and presents some taxonomic housekeeping for a monotypic taxon. Figures are relevant and useful. References are provided.

Experimental design

Experimental design is not relevant to this study. Information regarding the material examined is provided which would allow the specimens to be re-examined. This is consistent with standard taxonomic practice.

Validity of the findings

Statistics are not relevant. Conclusions appear valid although some mention of the characters used to differentiate the species in Friese's original study should be discussed and arguments made to refute their validity. The synonymy is subjective, but this is true of all such taxonomic decisions and follows standard practice.

Comments for the author

Review of Almeida and Quintero

The authors clarify the taxonomic status of a monotypic Neotropical bee Hoplocolletes ventralis (Friese). The type material was apparently mislabeled leading to some earlier confusion about the range of the species. The authors propose a synonym and describe the male for the first time. This is a straightforward case of some taxonomic housekeeping which also provides some useful information regarding the taxonomy of the bee. I see no issue with the publication being accepted once some minor edits have been made to the manuscript, which I list below.

Abstract.
Line 11. Neopasiphaeinae is used as an adjective, but this should be Neopasiphaeine bees instead. I also think “well” known is perhaps overstating the knowledge of its Amphinotic distribution. I expect the family is not very well known let alone its distribution.
Line 15. I would change sentence beginning with “Despite this proximity…” to read “No neopasiphaeine species occurs on both sides of the Pacific Ocean, but the Neotropical species Hoplocolletes ventralis (Friese, 1924) was described …”
Line 21. Replace Dasycolletes with Hoplocolletes. Up to this point there is nothing to indicate that Dasycolletes ventralis is the same species as Hoplocolletes ventralis.

Line 33. I’m not sure it can be assumed that the species to have been described from a single female. Friese simply uses the symbol ♀ without a number. It is true that he sometimes precedes this with a number such as in the description of Dasycolletes rufoaeneus, but he also does not use a number for the specimens in the description of D. chalceus although he provides a range of sizes, which implies or at least leaves open the possibility of multiple specimens in the type series. See Rasmussen & Ascher 2008: pg. 11, second last paragraph.

Line 39. Replace “considering its” with “based on”

Line 41. Replace “as” with “a”
Line 48. Replace “male undescribed for decades…” with “the male remaining undescribed, host-plant preferences are unknown”…
Line 50. Inconsistent use of abbreviations throughout (EAB Almeida vs. J.S.Ascher [line 100], see acknowledgements)
Line 56. Neophasiphaeinae used as an adjective again

Line 56. “since taxa with this character, Hoplocollete, Cephalocolletes…”
Line 65. Replace Dasycolletes ventralis with Hoplocolletes ventralis to avoid confusion
Line 100. Be sure to put spaces between first and last names “J. S. Ascher”.
Line 109/line 112. As stated above, a case could be made that there is no clear implication of the descriptions being based on a single specimen, especially in the case of D. chalceus in which the size is given as a range. ICZN recommendation 73F “avoidance of assumption of holotype” would seem to imply that these could be considered as syntypes. It might be worth considering designating a lectotype or given that the AMNH specimen may be a duplicate as stated above a neotype (or simply list these as syntypes).

Line 111. State clearly that this is a "new synonymy". It would be relevant to discuss the limited characters provided by Friese in the original description of D. ventralis to differenitate it from D. chalceus and on what basis these characters are believed to be insufficient for recognizing the species as distinct.

Line 120: paraocular misspelled
Line 125: metapostnotum underlined for unknown reason
Line 154: apex of scape does not reach upper margin of median ocellus in the male. It would be sensible to separate characters present in both sexes, from ones found only in females and from ones found only in males. There are numerous cases where the character is provided without indicating that it is true of a single sex. Grouping them into these three categories would help clarify this to the reader and avoid having to indicate that characters of the scopa are present only in the female (lines 157–158.).
Figure 1. Legend. The text does not match the figure: C is a close up of the metasoma, lateral view, D is the face, E is T4-T6 dorsal view, F is wing G is specimen labels. (line 238: indicate that this a syntype not just a specimen; line 239 A- dorsal habitus (use lower case since the colon implies you are in a single sentence. (same is true of other figure legends.

---

## Round 0.2 · accepted · Accept

· Academic Editor

Accept

Thank you for your revisions.